# Title Assessing Potentially Inappropriate Medications in Seniors: Differences between American Geriatrics Society and STOPP Criteria, and Preventing Adverse Drug Reactions

**DOI:** 10.3390/geriatrics5040068

**Published:** 2020-09-30

**Authors:** Roger E. Thomas, Leonard T. Nguyen

**Affiliations:** 1Department of Family Medicine, Cumming School of Medicine, University of Calgary, Calgary, AB T2N 4N1, Canada; 2Data Analyst, Albert Precision Laboratories, Alberta Health Services, Calgary, AB T2N 4N1, Canada; leonard.nguyen@aplabs.ca

**Keywords:** potentially inappropriate medications, potential prescribing omissions, adverse drug reactions (ADRs), preventing ADRs, Clinical Support Decision Systems (CDSS), deprescribing

## Abstract

Key problems for seniors are their exposure to “potentially inappropriate medications” and “potential medication omissions”, which place them at risk for moderate, severe, or fatal adverse drug reactions. This study of 82,935 first admissions to acute care hospitals in Calgary during 2013–2018 identified 294,160 Screening Tool of Older People’s Prescriptions (STOPP) potentially inappropriate medications (PIMs) (3.55/patient), 226,970 American Geriatric Society (AGS) Beers PIMs (2.74/patient), 59,396 START potential prescribing omissions (PPOs) (0.72/patient), and 85,288 STOPP PPOs (1.03/patient) for which a new prescription corrected the omission. This represents an overwhelming workload to prevent inappropriate prescriptions continuing during the hospitalisation and then deprescribe them judiciously. Limiting scrutiny to the most frequent PIMs and PPOs will identify many moderate, severe, or fatal risks of causing adverse drug reactions (ADRs) but to identify all PIMs or PPO involving moderate or severe risks of ADRs also involves searching lower in the frequency list of patients. Deciding whether to use the STOPP or AGS Beers PIM lists is an important issue in searching for ADRs, because the Pearson correlation coefficient for agreement between the STOPP and AGS Beers PIM totals in this study was 0.7051 (95% CI 0.7016 to 0.7085; *p* < 0.001). The combined lists include 289 individual PIM medications but STOPP and AGS have only 159 (55%) in common. The AGS Beers lists include medications used in the US and STOPP/START those used in Europe. The AGS authors recommend using both criteria. The ideal solution to the problem is to implement carefully constructed Clinical Decision Support Systems (CDSS) as in the SENATOR trial, then for an experienced pharmacist to focus on the key PIMs and PPOs likely to lead to moderate, severe, or fatal ADRs. The pharmacist and key decision makers on the services need to establish a collegial relationship to discuss frequently changing the medications that place the patients at risk. Then, the remaining PIMs and PPOs that relate to chronic disease management can be discussed by phone with the family physician using the discharge summary, which lists the medications for potential deprescribing.

## 1. Introduction

“Potentially inappropriate medications” (PIMs) and “potential prescribing omissions” (PPOs) are key problems for older patients with multiple illnesses. A systematic review of 62 studies (*n* = 1,854,698) according to the Screening Tool of Older People’s Prescriptions (STOPP) [1] criteria found 42.8% of community dwelling individuals and 51.8% of hospitalised patients ≥65 had ≥one PIM and according to the AGS Beers [2] criteria, 58% and 55.5%, respectively, and many patients had multiple PIMs and PPOs [3]. A key issue is how to prevent prescription of PIMs and PPOs in the first place and then to deprescribe existing PIMs and PPOs to reduce the risk of adverse drug events (ADRs). The US Veterans Affairs is the largest database that has been analysed for ADRs and for the period 2009–2016 individuals 60–69 had an ADR rate of 15%, those 70–79 13%, 80–89 11%, and ≥90 9%. Of all ADRs, 5% were rated as severe [4].

### 1.1. Literature Review: RCTs to Reduce PIMs and ADRs 

Compared to the many cross-sectional studies there are only five randomised controlled trials assessing hospitalised patients ≥65 using the STOPP criteria to reduce the number of PIMs and/or ADRs. They are reviewed here in ascending order of the complexity of their interventions and outcome measures (Table 1). In a study of 146 patients ≥75 in Belgium, in the intervention group, the inpatient geriatric consultation team applied STOPP criteria (39.7% of PIMs were discontinued) and in the control group, geriatricians not familiar with STOPP provided their usual care (19.3% of PIMs were discontinued) (OR 2.75 (95% CI 1.22, 6.24; *p* = 0.013). However, after one year, 38% of the discontinued PIMs had been restarted in the intervention group and 43% in the control (n.s.). The author concluded that the key problem was compliance of ward physicians with the geriatricians’ recommendations [5].

In a study of 359 patients (average age 82.7) in Israel, the pharmacist provided 245 STOPP recommendations for 125 residents (the chief physician accepted 84%) and 82 START recommendations for 65 residents (accepted 92.6%). After 12 months, the rate of PIMs in the intervention group was 22.5% and in the control was 54% (*p* < 0.001), and for PPOs it was 6.3% and 21.9%, respectively (*p* < 0.001). The author concluded the chief physician’s high rate of acceptance of the recommendations enabled the success of the project [6].

In a study of 400, patients in Ireland (median age 76) were randomised to usual pharmacist care or usual pharmacist care + assessment with the STOPP criteria. There were 193 recommendations for 111 patients and the attending physicians accepted 91% of the STOPP and 97% of the START recommendations. The total Medication Appropriateness Index score at admission for the control group was 722 and for the intervention, 688, and after six months, they were 610 and 454, respectively (*p <* 0.001). The largest changes in medication inappropriateness were medication not indicated, not effective, dose incorrect, drug–disease interaction, and incorrect duration (all *p* < 0.001). The author concluded the decrease in PIMs was balanced by correction of PPOs by new needed prescriptions and that polypharmacy was not necessarily a measure of prescribing appropriateness [7].

In a study of 732, patients in Ireland were randomised to usual pharmacist care (medication reconciliation and surveillance of prescription order sheets with written specific advice to prescribers) or usual pharmacist care + assessment with STOPP/START criteria and answering clarifying questions. The pharmacist made 451 recommendations for 233 patients and the attending physicians implemented 237/292 STOPP (81%) and 139/159 START (87.3%). There were 45 ADRs in the intervention group (of which 42 were moderate or severe) and 31 were assessed as definitely avoidable in 31 patients and possibly avoidable in 14 patients. There were 89 ADRs in the control group (of which 71 were assessed as moderate or severe) and 85 as definitely or possible avoidable. The author concluded that the success of the project was due to the high acceptance rate of the recommendations by the attending physicians, and that significantly lower ADR rates could be accomplished by a single assessment early in the admission of unselected acutely ill seniors [8]. 

The large SENATOR RCT with 1537 patients in six European countries was intended to provide software support so that attending physicians could avoid PIMs, PPOs, and ADRs. A trigger list of adverse events with defined clinical symptoms reflecting crises in major organ systems in older people was devised: Falls; new onset unsteady gait; acute kidney injury; symptomatic orthostatic hypotension; major serum electrolyte disturbance; symptomatic bradycardia; new-onset major constipation; acute bleeding; acute dyspepsia/nausea/vomiting; acute diarrhoea; acute delirium; symptomatic hypoglycemia; and unspecified adverse event not specified above (e.g., acute liver failure anaphylaxis). The 828 trigger events for ADRs were identified in the 1537 patients and were classified using the Hartwig and Siegel criteria [9] as mild (215, 26%), moderate (564, 68.1%), severe (41, 4.9%), or fatal (8, 1%). In a second procedure, 475 confirmed primary end points (ADRs) were confirmed in 379 (24.7%) patients: Mild (84, 17.7%), moderate (364, 76.6%), severe (24, 5.1%), and fatal (3, 0.6%). A Clinical Support Decision System (CDSS) was provided [10].

The clinician adherence to SENATOR software recommendations was disappointingly low at an average 15% [10] to 17% [11] across the sites in six countries. 

The authors concluded the project did not succeed because most CDSS recommendations had low clinical significance, staff were busy and hospital stays were short, staff were unwilling to change medications they had not prescribed, and that was the family physician’s responsibility [10,11,12]. 

### 1.2. Purpose of This Study

The purpose of this study was to identify, in a large cohort of 82,935 patients admitted to the four Calgary hospitals 2013–2018 for their first admission in that period, the correlations of STOPP and AGS Beers PIMs and START PPOs with subsequent rehospitalisation and death and provide data to motivate physicians to focus on and deprescribe the PIMs and PPOs with the highest correlations with these adverse outcomes.

## 2. Materials and Methods 

### 2.1. Study Design and Participants 

The database consists of the charts of patients 65 or older admitted to the four acute-care Calgary hospitals (Foothills Medical Centre, Rockyview General Hospital, Peter Lougheed Centre, and South Health Campus) and discharged between 1 March 2013 and 28 February 2018. Their first visit recorded in this period is the focus of this study. All their medications were entered as their usual dosage and “potentially inappropriate medications” (PIMs) were assessed using the criteria of the Screening Tool of Older People’s Prescriptions (STOPP) [1] and AGS Beers [2].

The Alberta Health Services’ Data Integration, Management, and Reporting database (DIMER) service accessed data from the Alberta Health Services (AHS) registration database and the Pharmaceutical Information Network (PIN) to provide anonymized admission and discharge records, medications, and laboratory data. 

### 2.2. Potentially Inappropriate Medications and Potential Prescribing Omissions

For each admission their diagnoses, co-morbidities, and admission and discharge medications provided data to apply 78/80 STOPP PIM and 28/34 START PPO criteria (2015 criteria). Due to lack of data we were not able to apply these STOPP criteria: Drugs prescribed without evidence-based clinical indication (A1) and prescribed beyond recommended duration (A2) and these START PPO criteria: Home continuous oxygen with chronic hypoxaemia (B3), fibre supplement for diverticulosis with constipation, annual influenza (I1) and pneumococcal (I2) vaccines, and vitamin D/calcium supplements for musculoskeletal issues (E2, E3, E5). We were able to assess 69 AGS Beers 2019 criteria but not PIMs affecting the renal system because laboratory data were not available.

Physicians could enter admission and discharge diagnoses and comorbidities in the electronic medical records (EMRs) in the four hospitals without using ICD-9 or -10 codes. Therefore, we constructed a lexicon simplifying the multiple ways in which the same diagnosis was entered. We did not create ICD-10 codes for each patient as we did not have Ethics permission to examine individual charts. Also, there were minimal ADR diagnosis categories in the hospitals’ EMRs, which resulted in a low rate of ADRs in which we did not have confidence and these rates are not reported here.

### 2.3. Similarities and Differences between the STOPP and AGS Beers PIM Lists

The STOPP and AGS Beers PIM criteria were compared to assess their similarities and differences. The STOPP and AGS Beers criteria publications list medication classes and STOPP mentions few medications by name although AGS does list more. It is thus necessary to complete the drug classes by adding the names of individual medications. Lists of complete medications were not available for the STOPP/START criteria from the Clanwilliam IT firm, which programmed the SENATOR trial or from the AGS office, so the drug class lists were completed using the Compendium of Pharmaceuticals and Specialties of the Canadian Pharmacists Association [13] and 289 individual PIM medications were derived. Of these, STOPP and AGS Beers had 159 (55%) in common. An additional difference is that although the STOPP and AGS Beers lists begin with anatomic classifications then therapeutic indications, they are structured differently.

### 2.4. Analysis

The statistical package R studio [14,15] was used to manage the dataset, and logistic regressions were computed to ascertain correlations between age, sex, numbers of medications on admission and discharge, numbers of PIMs and PPOs, individual PIMs and PPOs, and groups of PIMs and PPOs with the outcomes of rehospitalisation or death within six months of discharge. Six months was chosen to allow adverse effects of medications sufficient time to manifest as correlations with adverse events. 

### 2.5. Artificial Intelligence

In the current study, Association Rule Mining (ARM) [16,17,18] was used to identify both individual PIMs and PPOs and groups of PIM and PPO medications, which correlated with the outcomes of rehospitalisation or death within six months of discharge. The arules package’s apriori algorithm [19] assessed the 82,935 visits by grouping each PIM or PPO medication with other PIM or PPO medications in repeated iterations of medications datasets to identify correlations with the outcomes of rehospitalisation or death within six months of discharge, and each correlation was compared to the correlation assuming the outcomes and datasets were independent. The support threshold rule was set at 0.01 to limit the rules with low associations and required a PIM and an outcome must occur for at least 1% (829) of the patients. This resulted in 99 rules for AGS Beers PIM sets, 185 rules for STOPP PIM sets, 15 rules for START PPO sets, and 76 rules for START medications correctly prescribed for patients. The rules were ranked by the degree of lift (which measures the degree to which a PIM set is associated with an outcome compared to the situation in which events were completely independent) [20].

## 3. Results

### 3.1. Numbers of PIMs and PPOs

In this retrospective study of 82,935 patients, for their first admission to one of the four acute Calgary hospitals 2013–2018, there were 294,160 STOPP PIMs (3.55/patient) and 226,970 AGS Beers PIMs (2.74/patient). There were also 59,396 START PPOs (0.72/patient) and 85,288 STOPP PPOs (1.03/patient) for which a new prescription corrected the omission (Table 2). This study provides the most comprehensive comparison of STOPP and AGS Beers PIMs in the literature to date [20].

For PIMs with a count of more than 100 patients (100 patients comprise 0.12% of the study population) there were 58 STOPP and 46 AGS Beers PIM criteria, and 24 START PPOS and 22 START PPO criteria (for which physicians provided new needed prescriptions while the patient was in hospital). Even a list restricted to identifying individual PIMs and PPOs in groups of 100 patient or larger generates 150 PIM and PPOs events for attention. A further complication is that to identify many of the PIMs and PPOs that would constitute moderate or severe risks of ADRs involves searching lower in the frequency list of patients. The Pearson correlation coefficient for agreement between the STOPP and AGS Beers PIM totals in this study was 0.7051 (95% CI 0.7016 to 0.7085; *p* < 0.001). 

### 3.2. Correlations of PIMs and PPOs with Readmissions and Mortality

There was an increased risk of readmissions within a period of six months after hospital according to the number of medications individual patients took (OR = 1.09, 95% CI 1.09–1.09, *p* < 0.001); the number of STOPP PIMs (OR = 1.15, 95% CI 1.14–1.15 *p* < 0.001); the number of AGS Beers PIMs (OR = 1.15, 95% CI 1.14–1.16, *p* < 0.001); the number of START PPOs (OR = 1.04, 95% CI 1.02–1.06, *p* < 0.001); and the number of START PPOs corrected by prescriptions (OR = 1.16, 95% CI 1.14–1.17, *p* < 0.001) [20].

There was also an increased risk of death within a period of six months after hospital discharge according to the number of medications individual patients took (OR = 1.02, 95% CI 1.01–1.02, *p <* 0.001); the number of STOPP PIMS (OR = 1.07, 95% CI 1.06–1.08; *p <* 0.001); the number of AGS Beers PIMs (OR = 1.11, 95% CI 1.10–1.12, *p <* 0.001) and the number of START PPOs (OR = 1.31, 95% CI 1.27–1.34, *p <* 0.001). The number of PPOs corrected by prescriptions correlated with a minimal decrease in mortality (OR = 0.97, 95% CI 0.94–0.99, *p <* 0.0035) [20].

## 4. Discussion

The very large number of PIMs, PPOs, and corrected PPOs that occurred over a five-year period represents a potentially very heavy workload for a hospital pharmacist to assess and then work with attending physicians to correct. Hospital pharmacies are very busy, medications are often needed immediately on multiple services, and often have to be prepared from ingredients taking account of patient characteristics such as renal function, age, and weight. There is also the administrative burden of ordering medications and complex record keeping for many patients on multiple services. Moreover, this study reports only the first admission of each patient during the five-year period and the maximum number of admissions for a single patient during this period was 31.

There are several approaches to solving the problem of identifying the PIMS and PPOs most likely to cause moderate or severe ADRs.

### 4.1. Prioritising the Ten Most Frequent PIMS and PPOs

The pharmacist, while providing a comprehensive pharmaceutical assessment, could focus on the 10 most frequent PIMs by searching in each patient’s list for the 10 most frequent PIMs identified by summarising PIMs for the entire hospital. However, this approach presents several problems. 

#### 4.1.1. The Top Ten PIMS May Not Include PIMs with the Highest Risk of ADRs

The STOPP PIM top 10 list in this study includes several PIMs that are not at moderate, severe, or fatal risk of causing immediate ADRs. The STOPP PIMs with the highest numbers of occurrences were: Vasodilators with persistent postural hypotension (56,396), duplicate drug class prescriptions (49,949), regular opioids without laxative (25,880), aspirin, clopidogrel, dipyridamole, vitamin K antagonist, or thrombin/Factor Xa inhibitor with concurrent bleeding risk (17,350), strong opioid as first line therapy for mild pain (16,556), hypnotic Z-drugs (13,739), NSAID with severe hypertension (13,630), benzodiazepines (8667), β-blockers in diabetes mellitus with frequent hypoglycemia (8637), and loop diuretics as first-line treatment of hypertension (7431).

#### 4.1.2. The Ten AGS Beers Most Frequent PIMs in This Study Differ Substantially from the STOPP Top Ten List

Although nearly all the same individual medications are listed in STOPP and AGS Beers, the criteria differ because they were derived using different literature searches and different review groups, which used Delphi techniques. Moreover, AGS Beers includes medications used in the US and the STOPP/START medications in Europe. The AGS authors do recommend applying both sets of criteria. Thus, if the top 10 are focused on, they need to be prioritised according to the risk of causing moderate, severe, or fatal ADRs. 

#### 4.1.3. Identify Medications for Individual Patients That Are at Risk of Causing Moderate, Severe, or Fatal ADRs

The Senator trial demonstrated that their computer CDSS detected a high rate (24.7%) of ADRs, much higher than in previous studies and that the software worked well [10]. If the SENATOR RCT had been able to fund face-to-face or phone consultations between the pharmacists and the attending physicians and then with the family physicians, it would likely have been very successful. The trial was also based on comprehensive geriatric assessments of their older patients, an essential clinical element in understanding the diversity and complexity of individual older patients.

The pharmacist is the ideal health professional to implement medication changes because attending physicians and residents are often involved in emergencies or long ward rounds with other health professionals with these complex patients. The pharmacist needs to be provided with the authority and enough time to:(1)Make a prioritised list of the few essential medication changes to be made promptly in hospital likely to avoid moderate, severe, or fatal ADRs,(2)Agree by face-to-face or phone contact with the attending physicians to amend the drug ordering sheet after their conversation. The attending physician should expect to receive visits or calls from the pharmacists call during the clinicians’ ward activities, agree on decisions, and authorise pharmacists to change medication orders. Ideally, the same pharmacist and consultants should work together so they build up a solid and trusting working relationship.(3)The patient and carer should discuss with the attending physicians and pharmacist which other PIM, PPO, and ADR avoidance recommendations they would mutually like to resolve in hospital. If it is agreed they are best resolved by the family physician, it needs then to be agreed they can safely be deferred to the family physician. Pharmacists in Sweden in a non-randomised study of 400 hospitalised patients early in the admissions undertook detailed discussions with patients and their families about their medications and attitudes to medications and identified PIMs and PPOs. In this cohort, 12 months later, there were significant declines in the numbers of PIMs and PPOs, emergency visits assessed related to medications by 47%, and admissions related to medications by 80% [21,22].(4)The primary care physician should be contacted by phone and also receive a prioritised list of recommended changes, indicating that this has been discussed with the patient. An example of a change that could be deferred to the family physician could be low dose benzodiazepines without a history of falls. The family physician would then need to discuss with the patient other therapies to resolve the patient’s anxiety issues. Without a discussion that satisfies the patients, medications may be restarted. Dalleur’s study showed that at 12 months 38% of discontinued PIMS were restarted in the intervention and 43% in the control group [5].

### 4.2. Identify at the Health System Level Medications for All Patients That Are at Risk of Causing Moderate, Severe, or Fatal ADRs 

There also needs to be active and prompt oversight at the health system level to identify medications likely to cause moderate, severe, or fatal ADRs. This can be done by continuously updating the ADR list and identifying the medications involved at the health-system level. This could also be accomplished by artificial intelligence (AI) using, for example, Association Rule Mining, which iteratively compares lists of medications for ADR risks. 

The ARM approach would identify, at the system level, specific medications and combinations at especially high risk of causing moderate, severe, or fatal ADRs and could be the basis of strong recommendations in the Clinical Advice Tool in the ESR, or actions to be taken by the pharmacy service to discontinue specific medications.

## 5. Conclusions

### 5.1. The Problem of Multiple PIMs and PPOs: Should They All Be Resolved during the Period of Hospitalisation?

This retrospective study of 82,935 first admissions of individuals ≥ 65 to the four acute care hospitals in Calgary, Alberta 2013–2018 showed high levels of PIMs and PPOs. Over the five-year period there were 294,160 STOPP PIMs (3.55/patient) and 226,970 AGS Beers PIMs (2.74/patient), 59,396 START PPOs (0.72/patient) and 85,288 STOPP PPOs (1.03/patient) for which a new prescription corrected the omission (Table 2). This study provides the most comprehensive comparison of STOPP and AGS Beers PIMs in the literature to date. The study demonstrates that inappropriate prescribing presents four interrelated problems requiring resolution [20]: (1) This is a huge workload for the pharmacists and physicians; (2) this study demonstrated that pharmacists would typically encounter at least 150 types of PIMs and 24 of PPOs; (3) the numbers of individual prescriptions to be corrected over a five-year period in these four hospital is very large: STOPP PIMs with the highest numbers of occurrences were: Vasodilators with persistent postural hypotension (56,396), duplicate drug class prescriptions (49,949), regular opioids without laxative (25,880), aspirin, clopidogrel, dipyridamole, vitamin K antagonist, or thrombin/Factor Xa inhibitor with concurrent bleeding risk (17,350), strong opioid as first line therapy for mild pain (16,556), hypnotic Z-drugs (13,739), NSAID with severe hypertension (13,630), benzodiazepines (8667), β-blockers in diabetes mellitus with frequent hypoglycemia (8637), and loop diuretics as first-line treatment of hypertension (7431). Because the STOPP and AGS Beers lists only had a Pearson correlation of 0.7051 it would be prudent to combine them.

### 5.2. The Solution: Identify Key PIMs and PPOs with Risk of Moderate, Severe or Fatal ADRs and Resolve Then through Discussion by the Pharmacist and Key Decision Makers on the Services

The solution to the problem is to implement carefully constructed CDSS as in the SENATOR trial, then for an experienced pharmacist to focus on the key PIMs and PPOs likely to lead to moderate, severe, or fatal ADRs. The pharmacist and key decision makers on the services need to establish a collegial relationship to frequently discuss changing the medications that place the patient at risk. Then, the remaining PIMs and PPOs that relate to chronic disease management can be discussed by phone with the family physician using the discharge summary, which lists the medications for potential deprescribing.

## Figures and Tables

**Table 1 geriatrics-05-00068-t001:** Randomised controlled trials of assessing Screening Tool of Older People’s Prescriptions (STOPP) potentially inappropriate medications (PIMs), START potential medication omissions (PPOs), and adverse drug reactions (ADRs) in older hospitalised people.

Author, Date, Country, *n*, % Female, Median Age	Method of Selecting Patients for Assessment of ADRs and Method of Randomisation	Median no Illnesses, Median Meds Admission and Discharge	STOPP and ADR Measures	Results, Conclusions and Recommendations
Dalleur 2014 [5], Belgium, 146; 63% female; 85 years	Inclusion criteria: ≥75 years, risk of frailty defined by Identification of Seniors At Risk ^1^ score ≥2/6, admission to a medical ward, comprehensive geriatric assessment (CGA) ^2^ confirming frailty performed by inpatient geriatric consultation team (IGCT); patients randomised by nurse drawing lots, 2 geriatricians familiar with STOPP assigned to intervention ICGT, 2 geriatricians who had never worked with STOPP assigned to control	Median meds 7; 82% polypharmacy (≥5 meds); 52% ≥ 1 PIM	(1) 64 STOPP criteria applied by intervention IGCT geriatricians; (2) 1 year later a geriatrician, GOP and clinical pharmacist evaluated STOPP recommendations to discontinue PIMS as 1. minor (no benefit or minor benefit); 2. moderate (improvement of the appropriateness of the level of practice or prevention of an ADR); 3. major (prevention of serious morbidity—including readmission—and serious ADR); 4. extreme (life-saving); 5 deleterious (increased risk of adverse health event); 6. Non-applicable	(1) 125 PIMs (41 benzodiazepines, 19 anti-platelet agents, 13 opiates, 10 β-blockers, 9 tricyclic antidepressants, 8 neuroleptics) (2) 39.7% PIMs discontinued in intervention, 19.3% in control group (OR 2.75 (1.22, 6.24; *p* = 0.013); 5 PIMs needed screening and advice to discontinue to achieve 1 PIM discontinued at discharge. (3) At 1 year, 38% of discontinued PIMs were restarted in intervention, 43% in control (ns) **Conclusion:** low compliance of hospital physicians with geriatrician recommendations is key problem
Frankenthal 2014 [6], Israel 359; 70.5% female; average age 82.7	Pharmacist (groups concealed from pharmacist) used sealed envelopes to randomise to ADL-dependent, ADL-independent and cognitively impaired groups; pharmacist conducted medication review at admission and 6 and 12 months, discussed with chief physician at admission and 6 months	Average 2.5 comorbidities; Medications: Intervention baseline 8.8 ± 3.4, control 8.2 ± 3; PIMs intervention 129 (70.5%), control 114 (64.7%); PPOs intervention 65 (35.5%), control 57 (32.4%)	(1) Randomised to receive STOPP assessment vs. usual pharmaceutical care, (2) Other outcome measures: Medical Outcomes Study 12-item Short-Form Health Survey (for falls and hospitalisations), Functional Independence Measure (13 motor and 5 cognitive items on 7-point scale, range 18 (total dependence) to 126 (independence)	(1) Pharmacist gave 245 STOPP recommendations for 125 residents to chief physician (84% accepted); and 82 START recommendations for 65 residents (92.6% accepted) (2) Mean no. medications at 12 months intervention 7.3 ± 2.7, control 8.9 ± 3.2 (*p* < 0.001) (3) PIMS 12 months intervention 36 (22.5%) control 79 (54%) (*p* < 0.001); 43 individual PIM medications identified. Those affecting ≥ 10% of patients were: diphenoxylate, loperamide or codeine for diarrhoea of unknown cause (19% of patients), duplicate drug prescriptions (16%), and prolonged use of first-generation antihistamines (16%) (4) PPOs 12 months intervention 10 (6.3%) control 32 (21.9%) (*p* < 0.001). 14 individual PPOs were identified (5) Decreased falls in intervention vs. control at 12 months (*p* = 0.006) (6) STOPP/START reviews took 5 minutes, reviews with chief physician median 20 min. **Conclusion:** Key factor in success was high rate of acceptance of recommendations by chief physician
Gallagher 2011 [7], Ireland, *n* = 400, median age 76, 53% female	Randomised to usual pharmacist care vs. usual pharmacist care + STOPP/START criteria	Average 2 comorbidities; Median 7.5 meds, MAI score intervention 8 (IQR 3–17.8), control 10 (IQR 3–16.3); AOU score intervention 37.5%, control 35.8%	Assessment of Underutilization of Medication Index (AOU)	(1) 183 recommendations for 111 (58.4%) patients to attending physicians; and 101 (91%) STOPP, 69 (97%) START recommendations accepted. (2) STOPP PIMs were cardiovascular 21, central nervous system 10, gastrointestinal 35, musculoskeletal 6, medications affecting falls 18, opiates 4, duplicate class prescriptions (3) START PPOS were cardiovascular system 40, respiratory 6, musculoskeletal 12, endocrine 11. (4) Total MAI scores admission: control 722 (47%) and intervention 688 (49%); at 6 months control 610 (41.8%) and intervention 454 (32.6%) *p* < 0.001, largest changes due to decreases in not indicated, not effective, dose incorrect, drug-disease interaction, incorrect duration (all <0.001) (5) No statistically different changes in falls, or rehospitalizations or deaths by 6 months **Conclusion:** reduction in PIMS counterbalanced by PPOs prescribed so polypharmacy reduction is not necessarily a measure of prescribing appropriateness.
O’Connor 2016 [8], Ireland, *n* = 732, median age 78, % female intervention 64%, control. 50%	Randomised to usual pharmacist care (medication reconciliation, surveillance of prescription order sheets with written specific advice to prescribers vs, usual pharmacist care plus STOPP/START criteria. Two clusters were identified in which consultants formed an integrated service. The orthopaedics, endocrinology, respiratory, renal, cardiology and radiation oncology services were assigned to the intervention group, and the general surgery gastroenterology, infectious diseases, respiratory, renal, cardiology and neurology services were assigned to the control group. Although respiratory, renal, and cardiology were represented in both clusters no consultant had patients in both clusters.	Average Charlson comorbidity Index 2; Barthel Index 18 (range 13–20); median prescription drugs intervention 9 (IQR 6–11), control 8 (IQR 6–11)	The primary researcher; (1) assessed STOPP/START criteria within 48 h of admission and reported PIMs and PPOs to resident or consultant and answered clarifying questions and placed printed report on patient’s record, (2) Applied WHO definition of ADR: response to a drug that is noxious and unintended and occurs at doses normally used for the prophylaxis, diagnosis or therapy of disease or for the modification of physiological function. (3) Applied WHO Uppsala Monitoring Centre criteria for probable or definite ADR: 1. Clinical effects consistent with the known side-effect profile of the drug according to the British National Formulary data, 2. a clear temporal relationship between the suspected ADR symptoms and initiation of drug (other causes of the adverse clinical symptoms and signs being excluded or highly unlikely), 3. affected individuals with one or more symptoms or signs defined according to a tigger list of the most-common clinical phenomena representing ADRs, 4. Independent corroboration by researcher blinded to group assignment using WHO Uppsala monitoring criteria. (4) assessed ADRs as (a). Moderate if caused hospital stay of >24 h beyond expected discharge date, significant deterioration in vital signs (blood pressure, heart rate, oxygen saturation, core temperature), or required specific corrective interventions; (b). Severe if directly caused death or permanent disability, necessitated admission to high-dependency unit or intensive therapy unit, or urgent administration of antidote.	(1) 451 recommendations for 233 participants: 292 STOPP (attending physicians implemented 237 = 81%), 159 START (139 = 87.4% implemented) (2) 45 ADRs in 42 (11.7%) patients in intervention group, definitely avoidable in 31 and possibly avoidable in 14 patients; 42 (93%) moderate or severe (3) 89 ADRs in 78 (21%) control group, 85 definitely or possibly avoidable and 71 (79.8%) moderate or severe; OR for ADR in intervention 0.50 (0.33, 0.75; *p* = 0.001) compared to control with absolute risk reduction (ARR) = 9.3%. (4) for intervention compared to control intervention 45 vs. 89 ADRs (7 vs. 19 opioids, 8 vs. 14 diuretics, 8 vs. 12 antihypertensives, 4 vs. 12 benzodiazepines, 5 vs. 8 ACEs or ARBs, 4 vs. 4 antibiotics, 5 vs. 8 anticoagulants, 3 vs. 8 anticoagulants, 3 vs. 8 nonsteroidals, 1 vs. 6 antiplatelets **Conclusions:** There was a high rate of acceptance of the recommendations. Application of STOPP/START criteria at an early single time in the hospitalisation of older people with acute unselected illnesses results in significantly lower ADR rates.
O’Mahony 2020 [10], 6 European countries, *n* = 1537, median age 78 (IQR 72, 84), 47% female,	large academic teaching hospital in each of 6 countries, patients well matched on age, sex, daily prescription drugs, CIRS-G, MMSE, BI scores and dependency level.	Barthel Index 18 (IQR 14, 20), MMSE median 27 (IQR 23, 29), daily medications 10 (IQR 8, 13), previous documented ADRs 669 (43.5%); falls previous 12 months 570 (37.1%), domestic assistance required (39.9%), personal care required (25.3%); smoker 108 (7%)	Trigger list of events: falls; new onset unsteady gait; acute kidney injury; symptomatic orthostatic hypotension; major serum electrolyte disturbance; symptomatic bradycardia; new-onset major constipation; acute bleeding; acute dyspepsia/nausea/vomiting; acute diarrhoea; acute delirium; symptomatic hypoglycaemia; unspecified adverse event not specified above (e.g., acute liver failure or anaphylaxis)	(1) 828 trigger events including 475 confirmed primary endpoints in 379 (24.7%) patients: 84 (17.7%) mild ADRs, 364 (76.6%) moderate, 24 (5.1%) severe, and 3 (0.6%) fatal; 190 control, 189 patients, OR 0.98 (0.77, 1.24; *p* = 0.88). (2) Uptake of advice by 15% of physicians 15% (no difference intervention and control) **Conclusions:** 1. Computerized advice reports frequently produced recommendations of low clinical significance in context of serious acute illness; 2. Busy pressurized acute hospital environment had negative impact on timing and location of medication advice delivery; 3. Prescribers had variable levels of experience/responsibility and attitude to clinical trials; 4. Clinicians’ variable knowledge of patients’ diagnostic details, medication preferences and clinical status in hospital; 5. Physicians’ belief long-term prescribing is responsibility of the patients’ primary care physician; 5. Reluctance to adjust medications outside one’s own expertise; and 6. Lack of awareness of highly prevalent ADRs and the high risk of incident ADRs in multi-morbid older patients; 7. short median length of stay of 6 days. **Recommendation:** It is essential to combine efficient software delivery of pharmacological advice with face-to-face contact with attending clinicians to promote comprehensive geriatric assessment and pharmacotherapy optimisation otherwise ADRs will continue to compromise patient safety.

^1^ Point for each of: Needing help with activities of daily life, increase in this need related to current illness, memory problems, significantly altered vision, hospitalisation previous 6 months, daily use ≥ 3 medications at home. ^2^ Screening for geriatric syndromes: ≥2 falls past 6 months; polypharmacy (≥5 daily medications); cognitive impairment (known dementia or Mini-Mental State Examination [MMSE] ≤ 24/30); body mass index < 21 kg/m^2^ and/or mid-arm circumference < 23 cm; living alone; and functional dependency in activities of daily life (Katz score ≥ 9/24).

**Table 2 geriatrics-05-00068-t002:** Most frequent STOPP and American Geriatric Society (AGS) Beers PIMs, START PPO omissions, and PPO prescriptions with counts of 100 or more patients.

STOPP			AGS Beers		
Code	Description	Count	Code	Description	Count
K3	Vasodilator with persistent postural hypotension	56,396	4D	Antipsychotics, benzodiazepine receptor agonist hypnotics, antidepressants (SSRI, SNRI, TCA), opioids (if history of falls)	40,806
A3	Any duplicate drug class prescription, e.g., two concurrent NSAIDs, SSRIs, SNRIs, loop diuretics, ACE inhibitors, anticoagulants	49,949	2D1	Peripheral alpha-1 blocker for hypertension	36,273
L2	Regular opioid without concomitant laxative	25,880	5E	≥3 CNS-active drugs (TCA, SSRI, SNRI, antipsychotic, antiepileptic, benzodiazepine, nonbenzodiazepine, hypnotic Z-drug, opioid)	14,306
C3	Aspirin, clopidogrel, dipyridamole, vitK antagonist, or thrombin/factor Xa inhibitor with concurrent bleeding risk	17,350	2E6	Benzodiazepine receptor agonist hypnotic Z drugs (increase delirium, falls, fractures, emergency room visits, hospitalisations, motor vehicle crashes)	13,739
L1	Oral or transdermal strong opioid as first line therapy for mild pain	16,556	2G3	Proton-pump inhibitor >8 weeks unless high-risk patient (oral corticosteroids, NSAID, erosive or Barrett esophagitis or failure of H-2 receptor antagonists	10,470
K4	Hypnotic Z-drug	13,739	2H2	Non-COX-2 selective NSAID 9incfreaed risk acute renal injury, further decline in renal function	9235
H2	NSAID with severe hypertension	13,630	2E4	Benzodiazepine (increased risk falls, fractures	8667
K1	Benzodiazepine	8667	3B3	Falls/fractures with anticonvulsant, antipsychotic, benzodiazepine	6663
J3	Beta-blocker in DM with frequent hypoglycaemia	8637	2E1	Antipsychotic (increased risk falls, fractures)	5332
B6	Loop diuretic as first-line treatment for hypertension	7431	2E2	Opioid with gabapentin or pregabalin (Increased risk of severe sedation-related adverse events, including respiratory depression and death)	4874
C11	NSAID with concurrent antiplatelet without PPI prophylaxis	7366	5C	Opioid with benzodiazepine (increased risk of overdose)	4800
E4	NSAID if eGFR < 50 mL/min/1.73 m^2^	5731	5B	≥2 anticholinergics (risk of confusion, dry mouth, constipation, toxicity, delirium)	4558
K2	Neuroleptic	5704	5D	Systemic corticosteroid with NSAID (Increased risk peptic ulcer disease or gastrointestinal bleeding)	4482
D5	Benzodiazepine for ≥4 weeks	5579	5F	Insulin, sliding scale (Higher risk of hypoglycaemia without improvement in hyperglycaemia management)	4268
C10	NSAID with vitK antagonist or thrombin/factor Xa inhibitor	5298	2F5	Nitrofurantoin (Potential for pulmonary toxicity, hepatoxicity, and peripheral neuropathy, especially with long-term use)	3845
M	Concomitant use of ≥2 antimuscarinic/anticholinergic drugs	4558	2C	Peripheral alpha-1 blocker with loop diuretic (Increased risk of urinary incontinence in older women)	3751
L3	Long-acting opioid without short-acting opioid for break-through pain	4433	5I	Antidepressant (Increased risk of falls (all) and of fracture)	3690
B12	Aldosterone antagonist with concurrent potassium-conserving drugs without monitoring of serum potassium	2756	2E1	Aspirin at age ≥70 for primary prevention of CVD or colorectal cancer (Risk of major bleeding from aspirin increases markedly in older age. Several studies suggest lack of net benefit when used for primary prevention in older adult with cardiovascular risk factors, but evidence is not conclusive)	3502
J1	Sulphonylurea with long duration of action	2736	4A	Sulfonylurea (Chlorpropamide: prolonged half-life in older adults; can cause prolonged hypoglycaemia; causes SIADH Glimepiride and glyburide: higher risk of severe prolonged hypoglycaemia in older adults)	3214
D7	Anticholinergic/antimuscarinic to treat extra-pyramidal side effects of neuroleptic	2178	3D2	Urinary incontinence with oral/transdermal oestrogen or peripheral alpha-1 blocker in women (Lack of efficacy (oral oestrogen) and aggravation of incontinence (alpha-1 blockers)	3153
I1	Antimuscarinic with dementia, chronic cognitive impairment, narrow-angle glaucoma, or prostatism	2092	2G1	Metoclopramide unless gastroparesis (Can cause extrapyramidal effects, including tardive dyskinesia; risk may be greater in frail older adults and with prolonged exposure)	3003
D8	Anticholinergic/antimuscarinic with delirium or dementia	2068	3A1	Heart failure with NSAID, COX-2, diltiazem, verapamil, thiazolidinedione, cilostazol, or dronedarone (Potential to promote fluid retention and/or exacerbate heart failure (NSAIDs and COX-2 inhibitors, nondihydropyridine CCBs, thiazolidinediones); potential to increase mortality in older adults with heart failure (cilostazol and dronedarone)	2999
B10	Centrally. acting antihypertensive unless clear intolerance of, or lack of efficacy with, other classes of antihypertensives	1970	2F3	Oestrogen with or without progestin (lack of efficacy)	2953
G2	Systemic corticosteroid with moderate-severe COPD	1909	2H3	Indomethacin, ketorolac (Increased risk of gastrointestinal bleeding/peptic ulcer disease and acute kidney injury in older adults. Indomethacin is more likely than other NSAIDs to have adverse most adverse and CNS effects)	2820
D9	Neuroleptic antipsychotic in patients with behavioural and psychological symptoms of dementia unless severe and after failed treatments	1894	5Q	Warfarin with NSAID (Increased risk of bleeding)	2811
I2	Alpha-1 alpha blocker with orthostatic hypotension or micturition syncope	1609	3B1	Delirium with anticholinergic, antipsychotic, benzodiazepine (Avoid in older adults with or at high risk of delirium because of potential inducing or worsening delirium. Antipsychotics are associated with greater risk of cerebrovascular accident and mortality in persons with dementia)	2714
F2	PPI at full therapeutic dosage for >8 weeks	1604	2A1	First-generation antihistamine (Highly anticholinergic; clearance reduced with advanced age, tolerance develops when used as hypnotic; risk of confusion, dry mouth, constipation, toxicity)	2708
H8	NSAID with systemic corticosteroid without PPI prophylaxis	1596	2D8	Nifedipine (Potential for hypotension; risk of precipitating myocardial ischemia)	2526
C5	Aspirin with vitK antagonist or thrombin/factor Xa inhibitor with chronic atrial fibrillation	1461	3B2	Dementia/cognitive impairment with anticholinergic, benzodiazepine, H2-receptor antagonist (potential of inducing or worsening delirium)	2195
B8	Thiazide diuretic with significant hypokalaemia, hyponatraemia, hypercalcaemia, or history of gout	1156	2D2	Central alpha blocker (High risk of adverse CNS effects; may cause bradycardia and orthostatic hypotension; not recommended as routine treatment for hypertension)	1970
B3	Beta-blocker in combination with diltiazem or verapamil	1013	2H4	Skeletal muscle relaxant (Most muscle relaxants poorly tolerated by older adults because some have anticholinergic adverse effects, sedation, increased risk of fractures; effectiveness at dosages tolerated by older adults questionable	1405
E6	Metformin if eGFR < 30 mL/min/1.73 m^2^	923	2A3	Antispasmodic (Highly anticholinergic, uncertain effectiveness)	1325
C9	VitK antagonist or thrombin/factor Xa inhibitor for first pulmonary embolus	917	2D5	Digoxin with atrial fibrillation (safer and more effective alternatives for rate control supported by high-quality evidence)	1174
H1	Non-COX-2 selective NSAID with peptic ulcer disease or gastrointestinal bleeding, unless with PPI or H2 antagonist	840	5N	Warfarin with ciprofloxacin (Increased risk of bleeding)	1138
B1	Digoxin for heart failure with normal systolic ventricular function	808	4B	Dabigatran, rivaroxaban at age ≥75 (Increased risk of gastrointestinal bleeding compared with warfarin and reported rates with other direct oral anticoagulants when used for long-term treatment of VTE or atrial fibrillation in adults ≥75 years.)	880
C4	Aspirin plus clopidogrel as secondary stroke prevention, unless stent	745	2D6	Digoxin with heart failure (Use in heart failure: evidence for benefits and harms of digoxin is conflicting and of lower quality)	808
D11	ACE inhibitor with a history of persistent bradycardia, heart block, recurrent syncope or concurrent drugs that reduce heart rate	715	3A2	Syncope with AChEI, peripheral alpha-1 blocker, tertiary TCA, chlorpromazine (AChEIs cause bradycardia and should be avoided in older adults whose syncope may be due to bradycardia. Nonselective peripheral alpha-1 blockers cause orthostatic blood pressure changes and should be avoided in older adults whose syncope may be due to orthostatic hypotension. Tertiary TCAs and the antipsychotics listed increase the risk of orthostatic hypotension or bradycardia)	767
B4	Beta-blocker with bradycardia, type II or complete heart block	645	2D9	Amiodarone (Effective for maintaining sinus rhythm but has greater toxicities than other antiarrhythmics used in atrial fibrillation; may be reasonable first-line therapy in patients with concomitant heart failure or substantial left ventricular hypertrophy if rhythm control is preferred over rate control)	575
H5	Systemic corticosteroid for osteoarthritis	623	5M	Warfarin with amiodarone (Increased risk of bleeding)	572
C6	Antiplatelet with vitK antagonist, thrombin/factor Xa inhibitor with stable coronary, CV or peripheral arterial disease	500	5N	Warfarin with macrolide (Increased risk of bleeding but excluding Azithromycin)	442
F3	Antimuscarinic/anticholinergic, oral iron, opioid, verapamil, or aluminium antacid with chronic constipation	546	2F1	Androgen (Potential for cardiac problems; contraindicated in men with prostate cancer)	408
J5	Oral oestrogen without progestogen with intact uterus	435	3C	Gastric/duodenal ulcer with >325 mg/day aspirin or non-COX-2 selective NSAID (May exacerbate existing or cause new/additional ulcers)	383
B11	ACE inhibitor or ARB with hyperkalaemia	415	5A	Renin Angiotensin System (RAS) inhibitor, amiloride, or triamterene with another RAS inhibitor and CKD stage 3a or higher (Increased risk of hyperkalaemia)	329
J6	Androgen without hypogonadism	409	2D7	Digoxin at >0.125 mg/day (if used for atrial fibrillation or heart failure avoid dosages > 0.125 mg/day (moderate quality of evidence)	325
D12	Phenothiazine as first-line treatment	400	4E	Dextromethorphan/quinidine (Limited efficacy in treating patients with dementia symptoms disorder in absence of pseudobulbar affect while potentially increasing risk of falls and drug-drug interactions)	112
B2	Diltiazem or verapamil with NYHA Class III or IV heart failure	386			
H9	Oral bisphosphonate with upper gastrointestinal disease or bleeding, or peptic ulcer disease	373			
D4	SSRI with recent hyponatraemia	286			
E3	Factor Xa inhibitor if eGFR < 15 mL/min/1.73 m^2^	240			
C8	VitK antagonist or thrombin/factor Xa inhibitor for first DVT	236			
D1	TCA with dementia, narrow angle glaucoma, cardiac conduction abnormalities, prostatism, or urinary retention	229			
B13	PDE5 inhibitor in severe heart failure characterised by hypotension or concurrent nitrate therapy for angina	212			
D3	Neuroleptic with antimuscarinic/anticholinergic effects with history of prostatism or urinary retention	176			
D2	Initiation of TCA as first-line antidepressant treatment	146			
G4	Benzodiazepine with acute or chronic respiratory failure	122			
H7	COX-2 selective NSAID with cardiovascular disease	116			
	START Omission			START Inclusion	
H2	Laxative with regular opioids	25,471	A4	Antihypertensive therapy for hypertension	30,606
E4	Bone anti-resorptive or anabolic therapy with osteoporosis	5718	H2	Laxative with regular opioids	7053
A6	ACE inhibitor with systolic heart failure or coronary artery disease	4708	A1	VitK antagonist or thrombin/factor Xa inhibitor with chronic atrial fibrillation	6348
H1	High-potency opioid in moderate-severe pain, where paracetamol, NSAIDs or low-potency opioids are inappropriate or ineffective	2418	B1	Inhaled beta-2 agonist or antimuscarinic bronchodilator for mild/moderate asthma or COPD	5175
B2	Inhaled corticosteroid for moderate-severe asthma or COPD	2263	A6	ACE inhibitor with systolic heart failure or coronary artery disease	4570
A4	Antihypertensive therapy for hypertension	2222	A8	Appropriate beta-blocker with stable systolic heart failure	4441
A8	Appropriate beta-blocker with stable systolic heart failure	1928	B2	Inhaled corticosteroid for moderate-severe asthma or COPD	4433
G2	5-alpha reductase inhibitor with prostatism and no prostatectomy	1896	H1	High-potency opioid in moderate-severe pain, where paracetamol, NSAIDs or low-potency opioids are inappropriate or ineffective	3609
A1	VitK antagonist or thrombin/factor Xa inhibitor with chronic atrial fibrillation	1528	F	ACE inhibitor or ARB in diabetes with renal disease	2674
A2	Aspirin with chronic arial fibrillation and contraindicated VitK antagonist or thrombin/factor Xa inhibitor	1528	A5	Statin with coronary, cerebral or peripheral vascular disease	2673
B1	Inhaled beta-2 agonist or antimuscarinic bronchodilator for mild/moderate asthma or COPD	1521	A3	Antiplatelet with coronary, cerebral or peripheral vascular disease	2584
F	ACE inhibitor or ARB in diabetes with renal disease	1415	E4	Bone anti-resorptive or anabolic therapy with osteoporosis	2353
A3	Antiplatelet with coronary, cerebral or peripheral vascular disease	1301	D1	PPI with severe gastro-oesophageal reflux disease or peptic stricture	1667
G1	Alpha-1 receptor blocker with prostatism and no prostatectomy	998	C2	Non-TCA antidepressant with persistent major depressive symptoms	1249
C3	ACE inhibitor for mild-moderate Alzheimer’s or Lewy Body dementia	704	G1	Alpha-1 receptor blocker with prostatism and no prostatectomy	1190
C2	Non-TCA antidepressant with persistent major depressive symptoms	562	A2	Aspirin with chronic arial fibrillation and contraindicated VitK antagonist or thrombin/factor Xa inhibitor	934
C5	SSRI, SNRI or pregabalin for persistent severe anxiety	555	E7	Folic acid supplement with methotrexate	806
E6	Xanthine-oxidase inhibitor with recurrent gout	547	A7	Beta-blocker with ischaemic heart disease	687
A5	Statin with coronary, cerebral or peripheral vascular disease	529	C1	L-DOPA or dopamine agonist in Parkinson’s with functional impairment/disability	627
E1	Disease-modifying anti-rheumatic drug with active, disabling rheumatoid disease	424	C5	SSRI, SNRI or pregabalin for persistent severe anxiety	592
D1	PPI with severe gastro-oesophageal reflux disease or peptic stricture	366	E1	Disease-modifying anti-rheumatic drug with active, disabling rheumatoid disease	411
C1	L-DOPA or dopamine agonist in Parkinson’s with functional impairment/disability	299	C3	ACE inhibitor for mild-moderate Alzheimer’s or Lewy Body dementia	314
E7	Folic acid supplement with methotrexate	282	G2	5-alpha reductase inhibitor with prostatism and no prostatectomy	292
A7	Beta-blocker with ischaemic heart disease	163			

Notes. Alpha 1 antagonists: Prazosin, terazosin, cloxasosin are used for mild to moderate HTN but not monotherapy and other drugs classes are more effective in preventing heart failure. Major adverse effects are orthostatic hypotension and dizziness. They have high alpha 1 receptor affinity; alpha 2; phentolamine Alpha 1 = Alpha 2; labetolol, carvedilol β_1_ = ß_2_ ≥ alpha 1 > alpha 2 (Katzung 14th ed. p. 159). Sulphonylureas: First generation: Tolbutamide, chlorpropamide, Tolazamide; second generation: Glyburide, glipizide, gliclazide, glimepiride are 100–200 times more potent than tolbutamide so use with caution in elderly patients in whom hypoglycaemia would be especially dangerous. Glyburide contraindicated in hepatic impairment and renal insufficiency; glipizide is 90% metabolised in liver and is contraindicated in patients with significant hepatic impairment. Due to lower potency and shorter duration of action preferable to glyburide in elderly and with renal impairment. Glimepiride half-life 5–9 h and are completely metabolised in liver. (Katzung 14th ed. p. 758). Urinary incontinence: Anticholinergics/antimuscarinics: Oxybutynin, tolterodine, fesoterodine, trospium, darifenacin, solifenacin competitively block bladder 2 and 3 receptors, decrease detrusor muscle contractions, and relax bladder and reduce urge. All equally effective so choose on basis of AEs. vs. placebo 56% vs. 41% perceive cure or improvement, NNT = 7; 4 fewer leakage episodes and 5 fewer voids/week. Adverse Events are: Dry mouth is common reason to stop Rx (range 13–41%, severe 8%; OXY IR > OCY XL > darifenacin = solifenacin > TOLT IR > TOLT ER > Trospium > Trospium ER > Gelnique = OXY patch); blurred vision; constipation (trospium > oxybutynin 5 mg bid & darifenancin = solifenacin > tolterodine); to cope with dry mouth use sugarless candy, saliva substitutes (e.g., Oral Balance Gel, Mouth Kote, Biotene, or Moisir GI upset; GERD, dizziness (3%); HA; drowsiness; heat intolerance and pruritus. Patch-specific: Pruritus 17%, erythema 50%, less dry mouth, avoids first pass effect. Serious AEs: MVAs, decreased cognition, confusion (<1%), convulsions, falls, bradycardia, tachycardia, flushing, anxiety, allergy angioedema (tolterodine, darifenacin), QTc prolongation in at-risk pts (tolterodine, solifenacin); sweating (solifenancin); Mirabegron is a B3 agonist and increases bladder capacity, but increases HR, Bp, and QT; Adverse Effects are headache, constipation, fatigue.

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
