# Peer review of "Title Assessing Potentially Inappropriate Medications in Seniors: Differences between American Geriatrics Society and STOPP Criteria, and Preventing Adverse Drug Reactions"

_geriatrics, 2020, doi:10.3390/geriatrics5040068_

Round 1
Reviewer 1 Report
Specific Comments:
Abstract
Define all abbreviations throughout
Introduction
General comment: Entirely too much effort is put into reviweing existing literature on this topic. This needs to be greatly condensed. The current work should focus on this new original research, not provide enough detail for a review itself.
Line 44: quotes are unnecessary for abbreviations
Line 53: Abbreviate American Geriatrics Society as AGS when first mentioned
Material and Methods
Line 176: The criteria endorsed by the American Geriatrics Society is known as the Beers Criteria. It is unclear to me why it was not referred to as such in this manuscript. It is also copyrighted so ensure correct notation is used throughout.
Line 221: Why was 6 months used for the outcome assessment?
Discussion
Line 330: add a return to this line to separate text from heading
Conclusion
No need to present data again in this section.
Author Response
Reviewer Number 1
Many thanks for your helpful comments. They are appreciated.
- Abbreviations have been defined.
- I agree, the introduction is much too long and has been abbreviated. It is intended to motivate the purpose of this study and the introduction is now linked to the purpose of the study.
- Line 44: Quotes have been removed
- Line 53: American Geriatric Society (AGS) was written out in full in the Abstract and all subsequent references are to AGS
- Line 176. AGS criteria has now been amended to AGS Beers criteria and in all subsequent mentions
- Line 221: six months was chosen as a time period in which the correlations of PIMs and PPOs with the adverse events of rehospitalisation and death would have had time to be apparent. It is an arbitrary choice and has been used in other studies.
- Line 330: Return added
- Conclusions: I agree, no need to present data in full again and the section has been rewritten.
Reviewer 2 Report
I do not have any further comments or suggestions for the authors. Previous comments and suggestions have been addressed.
Author Response
Reviewer number 2
Many thanks for your comments and encouragement. Much appreciated! Roger Thomas.
Round 2
Reviewer 1 Report
No additional comments.
Author Response
I have already replied to the reviewer. I have now received recommendations from the Academic Editor.
I have completed those changes and will upload the revised manuscript, a COLOUR CODED revision which shows where the changes were made, and my reply to the Academic Editor.